# CAR NK Cell Therapy for the Treatment of Metastatic Melanoma: Potential & Prospects

**DOI:** 10.3390/cells12232750

**Published:** 2023-11-30

**Authors:** Winston Hibler, Glenn Merlino, Yanlin Yu

**Affiliations:** Laboratory of Cancer Biology and Genetics, Center for Cancer Research, National Cancer Institute, National Institutes of Health, Bethesda, MD 20892, USA

**Keywords:** natural killer cell, CAR-NK cell, melanoma, metastasis, CAR-NK cell therapy, cancer immunotherapy

## Abstract

Melanoma is among the most lethal forms of cancer, accounting for 80% of deaths despite comprising just 5% of skin cancer cases. Treatment options remain limited due to the genetic and epigenetic mechanisms associated with melanoma heterogeneity that underlie the rapid development of secondary drug resistance. For this reason, the development of novel treatments remains paramount to the improvement of patient outcomes. Although the advent of chimeric antigen receptor-expressing T (CAR-T) cell immunotherapies has led to many clinical successes for hematological malignancies, these treatments are limited in their utility by their immune-induced side effects and a high risk of systemic toxicities. CAR natural killer (CAR-NK) cell immunotherapies are a particularly promising alternative to CAR-T cell immunotherapies, as they offer a more favorable safety profile and have the capacity for fine-tuned cytotoxic activity. In this review, the discussion of the prospects and potential of CAR-NK cell immunotherapies touches upon the clinical contexts of melanoma, the immunobiology of NK cells, the immunosuppressive barriers preventing endogenous immune cells from eliminating tumors, and the structure and design of chimeric antigen receptors, then finishes with a series of proposed design innovations that could improve the efficacy CAR-NK cell immunotherapies in future studies.

## 1. Introduction

Melanoma, arising from the malignant transformation of melanocytes, is an aggressive and potentially lethal form of skin cancer. It is often caused by solar UV radiation, but there are myriad other environmental, immunological, and genetic risk factors that can also contribute to an individual’s predisposition towards melanoma [1,2]. For instance, heritable mutations such as in MC1R, CDK4, and CDKN2A, as well as genetic conditions like xeroderma pigmentosum (XP), have all been associated with an increased risk of melanoma development [3]. Melanoma is one of three major skin cancers, the other two being squamous cell carcinoma and basal cell carcinoma. Though melanoma accounts for only 5% of new skin cancer cases, it is ultimately responsible for approximately 80% of skin cancer deaths [4]. The incidence and mortality rates for malignant melanoma have steadily increased over the past few decades, particularly amongst older patients, whereas survival rates have remained relatively constant [4,5]. 

Resistance to a broad spectrum of therapies presents an ongoing challenge for the treatment of melanoma. Melanoma mutations occur at an exceptionally high rate, in turn driving the rapid development of treatment resistance [6]. Chemotherapeutics are often ineffective treatments for melanoma: standard chemotherapeutic agents such as dacarbazine elicit clinically positive responses in only 5–10% of patients [7,8]. If diagnosed early, melanoma could potentially be cured with surgery. However, roughly 30% of patients develop metastatic lesions in various organs following surgical ablation of primary tumors [9]. The propensity of melanoma for metastatic spread arises partly from the developmental origins of melanocytes, which are derived from the neural crest and consequently follow similar patterns of cellular migration to the brain and gastrointestinal tract [10]. Prognoses for patients with metastatic melanoma remain markedly poor. As the number of metastatic sites increases, the efficacy of treatments dwindles. Targeted therapies have improved patient outcomes; however, the effectiveness of these drugs is short-lived. For example, treatment with the BRAF inhibitors (BRAFi) vemurafenib and dabrafenib initially induced positive clinical responses in roughly 70% of BRAF-mutant patients [11], but the majority of responders subsequently developed BRAFi resistance in less than a year [12]. 

The 5-year survival rate for patients with metastatic melanoma is less than 5% [13] and highlights the limitations of current treatments. In recent years, the emergence of chimeric antigen receptor (CAR)-T cell therapies has garnered great success in the treatment of hematological cancers [14]. However, the application of CAR-T cell therapies is constrained by the associated risk of immunological systemic toxicities such as graft-versus-host disease (GvHD), immune effector cell-associated neurologic syndrome (ICANS), and cytokine release syndrome (CRS). GvHD and CRS tend to arise from immunoreactivity due to donor-host mismatching, whereas ICANS is an example of on-target, off-tumor toxicity. All of these can be fatal if left untreated [15,16,17,18,19]. More broadly, CAR-T cell therapies impose the risk of on-target, off-tumor toxicities that could cause potentially irreversible damage to healthy tissues that expressed the targeted antigen at low levels [20,21,22]. In addition, CAR-T cells are generally equipped with single-antigen specificity, so they are especially vulnerable to impaired efficacy in the event of antigen loss. Tumor heterogeneity is observed across many solid tumor types [23,24,25,26], including melanoma [27], which can further compromise the efficacy of CAR-T cell therapies if the targeted antigen is only expressed in a subset of tumor cells or becomes downregulated through acquired resistance [21,28,29].

CAR-NK cells are just beginning to gain traction as an alternative design to CAR-T cell therapies. Equipping natural killer (NK) cells with a CAR would retain the intrinsic cytotoxic capabilities of the transduced NK cell, while the phenotypic and mechanistic advantages of NK cells over T cells could allow CAR-NK cell therapies to soon become established as a viable treatment for melanoma. In contrast to T cells, adoptive cell therapies (ACTs) involving NK cells generally have a substantially lower risk of systemic toxicities. Furthermore, unlike T cells, NK cell activation does not require HLA matching for antigen recognition and its cytotoxicity mechanisms are regulated by multiple activating and inhibitory receptors [30,31,32]. This unique aspect of NK cell activity allows GvHD to be avoided in allogeneic NK cell therapies [31,33,34,35,36,37,38,39,40], which in turn enables allogeneic NK cells to be made readily available as an “off-the-shelf” treatment [41]. By extension, this implies that the acquisition of NK cells is generally much easier and more cost-efficient than T cells. Thus, CAR-NK cells have great potential as an immunotherapy treatment for melanoma.

## 2. Biology and Function of NK Cells in the Context of Melanoma

Accounting for 5–15% of the total cells in whole (peripheral) blood, NK cells are innate cytotoxic lymphocytes that can be found in blood, lung, liver, spleen, and bone marrow as well as in uterus, mucosal, and other tissues [42]. The two primary functions of NK cells are to maintain immune homeostasis and to eliminate stressed, infected, or potentially cancerous cells [43,44]. Generally, NK cells are characterized by the absence of CD3/TCR molecules and the presence of CD16 and CD56 (CD56^+^CD16^+^CD3^-^) phenotypes. Based on CD56 expression levels, NK cells can be further classified into CD56^dim^ and CD56^bright^ subsets [32]. CD56^dim^ NK cells account for 85–95% of the NK cell population within peripheral blood and co-express the Fcγ receptor CD16 that enables their participation as effector cells in antibody-dependent cell cytotoxicity (ADCC) [45,46]. The potent cytotoxic activity of the mature CD56^dim^ NK cells distinguishes them from the immature CD56^bright^ NK cells. CD56^bright^ NK cells account for only 5–15% of NK cells within peripheral blood, but, unlike CD56^dim^ NK cells, CD56^bright^ NK cells exhibit limited cytotoxic activity. Instead, CD56^bright^ NK cells predominantly function as immunoregulators by secreting cytokines and chemokines such as IL-12, IL-15, IL-18, IFNγ, and TNF-α that then recruit other immune effectors against target cells [47,48]. Such NK cell activities are precisely regulated by signals transferred through its receptors [32,44]. Structurally, NK cell receptors are divided into the immunoglobulin-like receptor (IgSF) superfamily and the C-type lectin-like receptor (CLSF) superfamily [48]. Functionally, NK cell receptors are classified as either activating or inhibitory receptors; both functional types are included within either superfamily [49,50]. The IgSF includes natural cytotoxicity receptors (NCRs), human killer cell immunoglobulin-like receptors (KIRs), and leukocyte immunoglobulin-like receptors (LILRs) [50]. Killer cell lectin-like receptors (KLRs) are the primary constituents of the CLSF [50].

### 2.1. NK Cell Activating Receptors

NK cell activation signals are mediated by a diverse group of receptors. The main NK cell activating receptors (NKARs) include NCRs, NKG2D, NKG2C, DNAM-1 (CD226), CD16, and activating KIRs (aKIRs). NCRs play a large role in NK cell-mediated cytotoxicity against malignant cells [51]. The three NCRs expressed by NK cells are NKp46 (NCR1), NKp44 (NCR2), and NKp30 (NCR3) [52]. NKp46 and NKp30 constitutively express in resting-state and activated NK cells, while NKp44 expresses restrictedly in activated NK cells [53]. Ongoing challenges in experimental designs leave many NCR ligands undiscovered, but a notable exception to this trend was the discovery of the B7-H6 protein as an activating ligand for NKp30 [54]. NKG2D is a CLSF NKAR expressed by all NK cells [52] and participates extensively in activation signal transduction. NKG2D can recognize a broad range of activating ligands including MICA, MICB, and UL16-binding proteins 1–6 (ULBP1-6) [55]. In addition, NKG2D is unusual in that it can mediate signal transduction via multiple possible signaling pathways [53,55]. MICA is an NKG2D ligand that has been reported to be strongly expressed in melanoma cells [56]. CD226 is an IgSF coactivating receptor that acts synergistically with other NKARs to induce NK cell activation. CD155 and CD112 are the two known CD226 ligands. CD112 is expressed in roughly 26% of melanoma cases, whereas CD155 is expressed in most primary and metastatic melanoma cases [57]. Although surface expression of CD155 can induce NK cell activation, melanoma cells secrete a soluble form of CD155 that inhibits CD226-mediated NK cell cytotoxicity [58], which may explain the correlation between CD155 expression and negative melanoma prognostic markers such as metastatic emergence and treatment resistance [59,60]. CD16 is also a member of the IgSF and similarly functions as an NKAR. Finally, aKIRs are IgSF receptors, and depending on how many Ig-like domains they possess, they can be further classified as either KIR2DS or KIR3DS, where the “**S**” denotes short cytoplasmic tails. aKIR are a subtype of HLA-I dependent KIRs with shorter intracellular domains and primarily participate in NK cell activation through the conduction of activation signals [61]. Individual aKIRs may vary in the particular HLA molecule that they recognize: KIR2DS1, KIR2DS2, and KIR2DS4 recognize HLA-C, while KIR3DS1 recognizes HLA-B [62].

Although CD16 binding to its IgG ligand alone is sufficient to induce the release of lytic granules from NK cells [63,64,65], individual NKARs generally cannot induce NK cell cytotoxicity alone; rather, NK cell cytotoxicity is initiated by the collective, synergistic signaling from multiple co-activated receptors. Activation signaling is influenced by the contributions of several other structures, many of which are formed from dimerization. One such example is the CD94:NKG2C heterodimeric complex, and much like aKIRs:DAP12, its primary functions involve the transmission of activating signals [62]. 

### 2.2. NK Cell Inhibitory Receptors

The chief inhibitory NK receptors include Inhibitory KIRs (iKIRs) and Killer cell lectin-like receptor D1(KLRD1). iKIRs typically have long cytoplasmic tails and account for the majority of all KIRs. iKIRs are classified with the same conventions as aKIRs. Long cytoplasmic tails are indicated by a trailing “**L**” to leave finalized categories of KIR2DL and KIR3DL. Despite having a long cytoplasmic tail, KIR2DL4 is one of the exceptions to the cytoplasmic tail trend due to its dual ability to transduce activating signals as well via its arginine–tyrosine activation motif [66]. Like aKIRs, iKIRs are also MHC-I dependent and are similarly varied in their HLA molecule specificity. KIR2DL1, KIR2DL2, and KIR2DL3 recognize various HLA-B and HLA-C allotypes. KIR3DL1 recognizes Bw4 epitopes among HLA-B and some HLA-A allotypes, while KIR3DL2 recognizes certain HLA-A allotypes [52]. Prognoses for metastatic melanoma have been reported to shift according to KIR expression. Lower expression of the aKIR KIR2DS5 corresponds to faster rates of melanoma progression [67]. Likewise, higher expression of the inhibitory KIR2DL2 receptor corresponds to greater melanoma progression [67]. An earlier study [68] reported that HLA-C-bound KIR2DL4 appeared to have protective roles in melanoma, where the KIR2DL3:HLA-C complex appeared to induce NK cell activation; the alternate explanation of spurious allelic variation has yet to be ruled out [69,70]. KLRD1 (CD94) heterodimerized with NKG2A (CD94:NKG2A) functions as an inhibitory receptor that has specificity towards HLA-E. HLA-E is typically expressed at negligible levels in melanoma cells, but surface expression of HLA-E can become drastically upregulated in the presence of IFNγ, which is produced in high quantities by NK cells [52]. The presence of NK cells could therefore induce HLA-E upregulation, and in turn, enable tumor resistance towards NK cell cytotoxicity [52]. This phenomenon is further corroborated by the restoration of cytotoxic efficacy via NKG2A inhibition, although this may result in IFNγ upregulation [71,72,73]. LIRs are IgR members that participate predominantly in inhibitory signal transduction. Examples of LIRs include LIR1 (ILT2), which notably demonstrates broad specificity to HLA-A, HLA-B, and HLA-G [62]. Like NKARs, inhibitory receptors can also transduce signals synergistically with coreceptors such as TACTILE (CD96) and TIGIT, the latter of which recognizes CD112, CD113, and CD155 [74,75] (Figure 1).

### 2.3. NK Cell Cytotoxicity and Significance to Tumor Cell Clearance

NK cell cytotoxicity can be activated through two different mechanisms: by secreting lytic granules containing perforin and granzymes to degrade vital cell structures of the target cell, or by expressing TRAIL or FasL on their cell surface to trigger apoptotic cascades in target cells when either ligand interacts with the corresponding receptor [76]. Strikingly, NK cell cytotoxicity is neither dependent on antigen recognition nor restricted to one particular antigen [77]. Instead, NK cell activities are mediated by an intricate balance of germline-encoded activating or inhibitory receptors that can elicit NK cell responses as needed yet still ensure self-tolerance. Though the absence of self-identifying surface HLA expression is generally necessary for NK cells to be able to activate their cytotoxicity, this alone is not sufficient. Attenuated inhibitory signals must also be accompanied by activating signals from receptor recognition of specific ligands. Stimulatory ligands are only upregulated in stressed cells, so under normal circumstances, the inhibitory effects of HLA-I expression by healthy cells combined with the absence of activating ligands ensures that NK cells remain tolerant toward them [78]. Malignantly transformed cells, on the other hand, tend to downregulate the surface expression of HLA-I and upregulate stress-induced activating ligands, which initiate NK cell cytotoxicity against cancerous cells [79].

NK cells fulfill such a critical role in tumor cell clearance that the degree of NK cell infiltration in tumors can be used as a heuristic predictor of patient outcomes. Infiltrated NK cells in melanoma tumors were found to be positively correlated with patient survival rates [80,81]. The findings from a later study described a negative correlation between patient survival rates and the proportion of weakly cytotoxic CD56^bright^ NK cells in circulation, which implicitly validated that the proportion of strongly cytotoxic CD56^dim^ NK cells in circulation or infiltrated into tumors indeed correlated positively with patient survival [82]. The occurrence of both tumorigenesis and metastatic invasion is contingent upon tumor cell escape of NK cell-mediated destruction [32]. As such, the development of novel therapies that can disrupt immune escape mechanisms is paramount to patient survival.

## 3. NK Cell Immunosuppression within the Tumor Microenvironment

The tumor microenvironment (TME) is an evolving, complex, and dynamic entity that is characterized by metabolic and chemical conditions that are unfavorable for an immunological response to the growing tumor. Ongoing crosstalk between melanoma cells, immune cells, and stromal cells within the TME gives rise to a variety of functional shifts across several different cell types. Different cell types may not necessarily shift in the same direction, potentially shifting in activity to work in opposition to one another; however, phenotypic shifts tend to favor pro-tumoral states and often involve secretory activity. Several pro-tumoral cell types within the TME are capable of altering NK cell cytotoxicity, and the assortment of cell types that are present have already been summarized excellently in previous publications [83,84,85,86]. Here, we will focus on the three TME cell types that tend to exert the greatest influence over NK cell activity: regulatory T (T_reg_) cells, cancer-associated fibroblasts (CAFs), and tumor-associated macrophages (TAMs) (Figure 2).

The combined efforts of melanoma cells and pro-tumoral non-malignant cells give rise to four main NK cell immunosuppressive mechanisms by which tumor cells evade NK cell cytotoxicity: (1) disrupting receptor-ligand interactions responsible for NK cell activation; (2) amplifying receptor-ligand interactions that inhibit NK cell activity; (3) evading NK cell cytotoxicity with the assistance of non-malignant TME cells; and (4) escaping immune detection due to the immunosuppressive effect of inhospitable microenvironmental conditions (Figure 3). 

First, melanoma cells can circumvent NK cell cytotoxicity by disrupting ligand-receptor interactions that drive NK cell activation. This may be accomplished through the melanoma-mediated downregulation of NKAR expression on NK cell surfaces, which effectively places a constraint on the strength of NK cell activation signals. For instance, melanoma cells highly express indoleamine-pyrrole 2,3-dioxygenase (IDO1) and prostaglandin E2 (PGE2), both of which have immunosuppressive activity. PGE2 can directly modulate NK cell receptor expression, whereas IDO1 catalyzes the production of L-kynurenine, which can then interact with NK cells and drive downregulation of major NKARs such as NKG2D, NKp30, and NKp44 [87,88,89,90,91,92,93]. Melanoma cells may also interfere with NK cell activation by downregulation of NKAR activating ligand expression or/and clearing away NKAR-activating ligands from melanoma cell surfaces via metalloprotease-mediated cleavage. In the presence of the BRAFi vemurafenib, BRAF^V600E^-mutant melanoma cells displayed strong downregulation of the NKAR ligands MICA, CD155, and B7-H6 [94], which may result in melanoma cell resistance towards NK cell cytotoxicity as a direct byproduct of acquired BRAFi resistance. The metalloproteases ADAM10, ADAM17, matrix metalloprotein (MMP)-25, and MMP-14 have been reported to be highly expressed in melanoma [95,96,97]. In addition, proteolytic shedding of the NKAR ligands MICA, ULBP2, and B7-H6 was often observed in melanoma samples [98]. Metastatic melanoma cells frequently shed NKAR-activating ligands such as the NKG2D ligands MICA and MICB [99]. The soluble ligands freed by metalloprotease-mediated cleavage can still interact with the binding sites of NKARs, thus inhibiting NKAR activation by obstructing surface-bound activating ligands from reaching their intended binding sites [100,101]. Higher concentrations of soluble ligands freed by proteolytic cleavage have been associated with advanced melanoma, which may be a factor underlying poor prognoses for individuals with late-stage or metastatic melanoma [97,102]. Inhibiting ligand cleavage sites attenuated ligand shedding and prevented metalloprotease recognition, thereby restoring NK cell cytotoxicity [97,103].

Second, melanoma cells can suppress NK cell cytotoxicity by promoting inhibitory NK cell receptor-ligand interactions. Tumor cells upregulate their surface expression of inhibitory ligands such as CD111 [104], CD112 [105,106], CD155 [106,107], CD200 [108], PD-L1 [109,110], HLA-E [111], and HLA-G [112,113]. Expression of non-classical HLA-G and HLA-E allotypes is observed across numerous cancers. HLA-E interacts with the inhibitory NK receptor complex CD94:NKG2A to suppress NK cell activity [114,115]. Likewise, non-classical HLA-G is recognized by LIR1 and similarly drives inhibitory signaling in NK cells upon binding [112]. Melanoma is also suspected to promote the upregulation of inhibitory receptors in NK cells [52]. The inhibitory receptors Tim-3 and TIGIT are expressed in both NK and T cells to regulate effector cell activity, and both were found to have immunosuppressive effects on NK cell activity [32,116]. A later study demonstrated that NK cell exhaustion was successfully reversed upon blockade of Tim-3 [117]. TIGIT recognizes CD111, CD112, CD155 (PVR), and PVRL2, where each interaction has immunosuppressive effects on NK cell activity [105,106]. TIGIT is inferred to directly suppress NK cell cytotoxicity based upon previous reports that TIGIT and CD155 have inhibitory effects on NK cell production of IFNγ [107]. Cancer cells, including melanoma cells, can also evade immune responses through exosome-mediated delivery of inhibitory receptor ligands to NK cells [118,119]. 

Third, non-malignant cells in the TME can assist with melanoma immune evasion. Depending on the cell type and environment, TME cells may intercept and inhibit NK cell activity directly via cell-cell interactions or by secreting immunosuppressive signaling molecules. For example, CAFs are fibroblasts within the TME that have been redirected to a pro-tumoral state resembling the phenotype of myofibroblasts [120] and secrete a variety of immunosuppressive cytokines, growth factors, and proteases that promote extracellular matrix (ECM) remodeling into a more favorable environment for tumor growth [121]. CAFs have a key role as front-line regulators of the anti-tumor activity from tumor-infiltrating immune cells [122]. Proteases secreted by CAFs further obstruct NK cell activity by participating in the proteolytic cleavage of NKAR ligands from melanoma cell surfaces [123]. T_reg_ cells in the TME can similarly secrete immunosuppressive signaling molecules that can potentially impair NK cell cytotoxicity [124]. CD39/CD73 ectonucleotidase activity cleaves ATP to produce adenosine, which has the two-fold immunosuppressive effect of promoting CD4+ T_reg_ cell maturation [125,126] and halting activity for most cytotoxic effector cells. For instance, adenosine directly inhibits NK cell infiltration and activation [127]. The elevated adenosine levels can also prevent macrophages from activation to target tumor cells [90]. IDO secreted by TAMs and MDSCs, as well as PGE2 produced from CAFs, promote the recruitment of T_reg_ cells; in turn, this may further lead to intensified immunosuppression. TAMs were originally tumor-infiltrating immune cells that were driven from the phagocytic M1 phenotype to the angiogenic M2 phenotype through cell-cell interactions with melanoma (Figure 2) [90,128]. Furthermore, the acidic conditions of the TME promote the expression of the M2 phenotype [129,130]. TAM-secreted molecules may also disrupt treatment mechanisms. For example, TAMs secrete interleukin 1β (IL-1β) in response to the acidic conditions, which then intensely promote angiogenesis, tumor cell migration, tumor cell proliferation, and metastasis [129,130,131]. Analogously, CAFs secrete arginase into the TME, which appears to impair the arginine-dependent tumor cell detection mechanisms in cytotoxic T cells [132]. In cases of advanced disease progression, metastasis-associated macrophages (MAMs) may begin to express membrane-bound TGF-β that further compromises NK cell anti-tumor immunity [133]. 

Finally, immune evasion of NK cells may arise due to the inhospitable biochemical or environmental conditions of the TME [134]. Melanoma tumors do not use oxidative phosphorylation and are therefore heavily reliant on glycolysis for energy. Acidic byproducts from glycolysis and fermentation start to accumulate until the TME becomes distinctly acidic relative to its surroundings [135]. Moreover, uncontrolled tumor cell proliferation depletes the surrounding oxygen levels and results in hypoxic local conditions [136]. NK cell activation mechanisms, metabolism, and proliferative capacity are all dysregulated by the acidic and hypoxic conditions within the TME [82,106,108,113,114,132,137,138].

Inducing the expression of a CAR into NK cells helps redirect them towards malignant cells that express the target antigen and contributes activation signaling to the NK cell, thereby improving the chances that the NK cell can be cytotoxic against tumor cells despite the immunosuppressive effects of the TME. Both CAR-T cells and CAR-NK cells display enhanced cytotoxic activation capabilities alongside greater specificity against TAMs than their ordinary lymphocyte counterparts, and depending on the design, CAR-T and CAR-NK cells may be engineered with additional modifications that further increase their efficacy. 

## 4. Car NK Cell Therapies to Treat Melanoma

### 4.1. Design & Production of CAR-NK Cells

CAR-NK cells generally express a CAR that mechanistically imitates a TCR activation signaling pathway; in most cases, CAR-NK cells are still created by transducing NK cells with the same CAR-encoding genes that would be used to create CAR-T cells. The CAR itself consists of four modular components: a single-chain variable fragment (scFv) domain, a hinge domain, a transmembrane domain, and one or multiple intracellular signaling domains. The extracellular scFv tumor antigen binding domain confers the specificity of CAR cells. This domain consists of a heavy (VH) and light (VL) variable chain fragment that is connected by a linker [139]. The location and abundance of the epitope, the order of the variable chain fragments, and the length of the linker must all be taken into consideration when designing the scFv domain [15]. The hinge domain links the scFv domain to the transmembrane domain and the hinge domain length influences the formation of immunological synapses as it facilitates access to tumor antigens. Membrane-distal tumor epitopes are more accessible with longer hinges, while membrane-proximal epitopes are better suited to shorter hinges [140]. The hinge domain of CAR moieties is often assembled using IgG-based hinges derived from IgG1, IgG2, and IgG4, often including the CH2 or CH3 IgG domains; other hinge domain constructs have made use of peptide sequences derived from CD28 or CD8α [140]. The transmembrane domain consists of a hydrophobic α-helix that anchors the scFv and hinge domains to the NK cell membrane [15,139]. CD4, CD28, and CD8α are the most commonly used transmembrane domains, but the choice of transmembrane domain may impact the degree of cell activation. The intracellular signaling domain governs the strength and nature of the activation signal. Differences in this intracellular domain distinguish between successive generations of CAR-transduced cells. First-generation CAR cells contained only the CD3ζ activation domain, while second- and third-generation CAR cells included one or two co-stimulatory molecules, respectively; of those molecules, CD28 and 4-1BB are frequently used [139,141]. Fourth-generation CAR cells are more broadly defined than the preceding generations but collectively seek to address the limitations of their predecessors.

Several fourth-generation CARs are constructed with a fragmented design. Universal CAR cells possess a fragmented and interchangeable antigen-specific extracellular domain that allows multiple cancer types or antigens to be targeted using the same combination of transmembrane and intracellular domains [15]. ON-switch CAR cells contain a fragmented intracellular domain that requires the presence of a particular small molecule for the functional CAR to be assembled; in other words, ON-switch CAR activation can be controlled through the administration of a drug [142]. Dual CARs comprise another subset of fourth-generation CARs; as the name implies, these constructs make use of two scFv domains simultaneously. AND-gated CAR cells require the expression of both target antigens on a given cell for signal propagation, which can enable non-specific tumor antigens to be targeted with better safety, provided that the other scFv domain is then targeted against a tumor-specific antigen. In contrast, OR-gated CAR cells activate when at least one of the two scFv domains has become bound to a target antigen to maximize tumor recognition rates. In both types of dual CARs, the scFv domains are able to share the same transmembrane and intracellular domains; in the case of OR-gated CARs, the dual scFv domains may simply involve the co-expression of two fully functional CAR constructs on the same cell [1]. Fourth-generation CARs also include a subset that can target the TME. In inhibitory CAR cells, the extracellular domains of inhibitory receptors are spliced together with the intracellular activating domains of CAR cells, allowing immunosuppressive signals to be redirected into activating signals [15]. Finally, T cells Redirected for Universal Cytokine Killing (TRUCK) are structurally similar to 3rd-generation CARs but carry an additional CAR-inducible transgenic “payload” that is to be delivered to the tumor site to attenuate immunosuppression by the TME [143]. Cytokine-mediated TME alteration is accompanied by a risk of systemic toxicities, so this approach should be used with caution, if at all [144]. 

### 4.2. Sources of NK Cells

NK cells suitable for CAR transduction can be obtained from several sources, but the process of isolating, purifying, and expanding primary NK cells can be difficult and inefficient. These limitations can be bypassed through NK cell lines, which are readily expanded in vitro [145]. The representative NK-92 cell line is an especially popular platform for many CAR-NK cell studies. Notably, CAR-NK92 cells were successfully used to treat renal cell carcinoma and high-risk rhabdomyosarcomas, and this may suggest that CAR-NK cells can elicit similar outcomes in additional cancer types [146,147]. One drawback of NK cell lines is their origin from malignant tissues, causing them to present an inherent risk of tumorigenicity [148]. As a precaution, NK cell lines must first be irradiated to prevent their persistence in vivo before they can be safely administered, but the shortened lifespan of irradiated cells may place a constraint on their clinical efficacy [149]. Transducing a drug-inducible suicide gene into CAR-NK cells may be an advantageous alternative to irradiation, as this would grant clinicians more agency in determining the duration of effect [150]. Umbilical cord blood is another viable source of NK cells. NK cells derived from cord blood possess an especially powerful capacity for expansion, giving them great clinical value in terms of scalability [151,152,153]. To minimize heterogeneity in the NK cell population between CD56^dim^ and CD56^bright^ phenotypes, cytokine co-stimulation ex vivo can induce the maturation into active CD56^dim^ effector NK cells that have appreciable cytotoxicity and durable persistence [154]. Peripheral blood is another source of NK cells rich in mature NK cells; however, despite the strong cytolytic activity and a tendency to persist in blood for longer periods, these NK cells are usually difficult to expand in vitro [154]. Peripheral blood NK cells are also at varying stages of maturation, which can result in an unfavorably heterogeneous cell population following expansion [154]. Finally, induced pluripotent stem cells (iPSCs) are another potential source of NK cells. Extraction of iPSCs from cord or peripheral blood yields an immature population that can be transduced with a CAR construct of interest. Subsequent incubation in a cytokine cocktail containing various interleukins and growth factors can then induce their differentiation into CAR-equipped NK cells [155]. This offers the benefit of a more homogenous population of CAR-NK cells, but at the cost of reduced cytotoxicity on account of a relatively larger proportion of immature cells [154].

### 4.3. Pre-Clinical CAR-NK Studies with Meaningful Implications for Melanoma

Numerous pre-clinical trials have studied CAR-NK cell therapies as a potential treatment for several different solid malignancies. Most of the existing pre-clinical data pertains to glioblastoma, breast cancer, ovarian cancer, and pancreatic cancer [154], but applications of CAR-NK cell therapies have currently been extended to colorectal cancer [155], neuroblastoma [156], hepatocellular carcinoma [157], gastric cancer [158], small cell lung cancer [159], and lung adenocarcinoma [160]. Importantly, multiple pre-clinical studies assess antigen targets that could be clinically relevant to melanoma (Table 1).

Pre-melanosome protein (PMEL; PMEL17; glycoprotein 100, gp100) ordinarily plays a role in melanin synthesis and is highly specific to melanoma [181]. Because PMEL is broadly overexpressed (61–90%) in melanoma patients [175,181,182], independent of their stage of disease progression, PMEL is of especially great interest as a therapeutic target. CAR-NK92MI cells bearing TCR-like CARs were assembled by selecting the TCR-like antibody GPA7 against the PMEL/HLA-A2 complex and then fused to the intracellular domain of a CD3-ζ chain [175]. In the context of HLA-A2, the anti-PMEL CAR-NK cells were able to recognize melanoma cells and exhibited enhanced cytotoxic activity. 

The disialoganglioside GD2 is overexpressed across multiple forms of cancer, such as glioblastoma [183], Ewing sarcoma [184], small cell lung cancer [185], melanoma [186], and other cancers, yet GD2 expression is generally restricted in healthy tissues [187]. Because of its strong cancer specificity, GD2 is a common target for several cancer immunotherapies [188]. GD2 is expressed by roughly 40% of melanoma tumors [182], and it follows that both CAR-T and CAR-NK cell therapies have been targeted against GD2 [172,173,189]. Mitwasi et al. generated anti-GD2 CAR-NK92 cells expressing CARs with either a conventional scFv domain or a novel IgG4-based antigen-binding domain and found that both CAR-NK92 variants demonstrated enhanced cytotoxic activity against melanoma cells both in vitro and in vivo; the IgG4-based CAR-NK92 variant exhibited a longer half-life than the conventional scFv-based variant but at the expense of reduced cytotoxic activity [172]. 

B7-H3 (CD276) is a transmembrane protein within the B7 glycoprotein family that is generally understood to be a negative regulator of effector cell activity [190,191]. Melanoma and several other solid tumors are known to express B7-H3, which may have an inhibitory effect on NK cell activity [192,193] and has grown to be a popular target for numerous immunotherapies, including both CAR-T cells and CAR-NK cells [161,194,195,196,197,198,199]. CAR-NK92 cells transduced with an anti-B7-H3 CAR exhibited strong cytotoxicity against melanoma cells independently of NKG2A expression or knockout, suggesting that anti-B7-H3 CAR-NK92 cells had displayed resistance towards NK inhibitory signaling mechanisms that are believed to underlie tumoral immune evasion [164]. Transforming growth factor (TGF)-β is another signaling molecule within the TME that has been associated with impaired cytotoxicity of both NK and CAR-NK cells, but this can be circumvented by co-transducing NK92-MI cells with both the anti-B7-H3 CAR and a TGF-β dominant negative receptor (DNR), thereby rescuing impaired cytotoxic activity [196].

Vascular endothelial growth factor receptor 2 (VEGFR-2) is an angiogenic receptor tyrosine kinase (RTK) that is upregulated in a subset of melanoma tumor specimens and is thought to play a role in tumor cell migration in metastasis [200]. In both T cells and YT NK cells, anti-VEGFR2 CAR transduction with an alternative design- replacing the scFv domain conventionally derived from monoclonal antibodies with an analogous fibronectin type III (Fn3) domain- successfully produced cytotoxicity against melanoma cells [201].

### 4.4. Potential Targets for CAR-NK Cell Therapies in Melanoma

Although melanoma has not yet been explicitly targeted in a CAR-NK clinical trial, there are a number of existing CAR-NK clinical trials that are currently recruiting or underway for various solid tumors. Several antigens from those studies may also apply to melanoma. A select few studies are discussed below, with more studies summarized in Table 2.

Mucin-1 (MUC1) is a transmembrane glycoprotein overexpressed in several solid tumors, such as breast cancer and melanoma. MUC1 could interact with ErbB2 and other RTKs involved in the PI3K/AKT signaling pathway, which in turn is associated with melanoma progression and acquired BRAF inhibitor resistance [201]. A recent study has shown that MUC1 participates in tumor cell migration as a driving force in metastasis in lung cancer, as well as melanoma [226]. A phase I clinical trial employed anti-PDL1/MUC1 dual CAR-NK cells against a range of solid tumors and found that this construct achieved a stable response in 9 out of 13 participants (~70%) [179]. By extension, this construct may be able to produce similar results if used to treat melanoma.

CLDN6 (cell adhesion protein Claudin-6), which plays a role in epithelial tight junction and cell-cell adhesion interactions, has recently been identified as a potential biomarker of melanoma and a myriad of other solid tumors [227]. A single-arm clinical trial (NCT05410717) is currently recruiting participants to investigate the therapeutic potential of anti-CLDN6 CAR-NK cells for the treatment of CLDN6-positive advanced solid tumors [166].

GD3 (ganglioside GD3) is in the same family of surface glycoproteins as GD2 and has been reported to be overexpressed in melanoma by at least 10-fold, relative to healthy melanocytes [228]. Upregulation of GD3 amplifies malignant invasion in melanoma cells by increasing adhesion signals and recruiting integrins [229,230]. Furthermore, GD3 was associated with malignant growth and invasion mediated by p130Cas and paxillin [229]. Previous pre-clinical studies have targeted GD3 as a melanoma antigen for both TCR-T and CAR-T cells [218,219], hinting that anti-GD3 CAR-NK cells could be used with similar success. Clinical implications for anti-GD3 CAR-NK cells may be particularly strong for melanoma brain metastases, which have been linked to GD3 upregulation [231], since CAR-NK cells offer an advantageous safety profile that can reduce the risk of on-target, off-tumor toxicities despite the tumor sites’ close proximity to ICANS-vulnerable brain regions.

αvβ3 (vitronectin receptor; integrin αvβ3) is expressed in melanoma, glioblastoma, pancreatic cancer, and other solid malignancies, where it plays a role in cell-cell and cell-ECM interactions [232]. Stimulation of integrin αvβ3 has led to activation of invasion by melanoma in vitro [233]. Alterations to patterns of glycosylation and sialylation of integrin αvβ3 have been reported in melanoma [234], hinting at its clinical significance as an antigen target. Anti-αvβ3 CAR-T cell therapies have been shown to effectively eliminate malignant cells in pre-clinical studies of glioblastoma, melanoma, and other solid malignancies [224,225]. The anti-α_v_β_3_ CAR construct can be translated to CAR-NK cells, where it would be expected to confer advantages such as improved safety and cost-efficiency over CAR-T cells.

CD126 (interleukin 6 receptor; IL-6R) is predominantly expressed in B cells and epithelial cells. Increased apoptosis and decreased malignant growth have both been reported after antibody-mediated inhibition of IL-6 signaling [235], which provided the basis for a pre-clinical CAR-T cell study that targeted CD126. Mishra et al. developed anti-CD126 CAR-T cells that demonstrated broad efficacy against a panel of cancer types, including metastatic melanoma, both in vitro and in vivo [205]. In light of this success, anti-CD126 CAR-NK cells may be able to improve upon the clinical viability of this CAR-T cell construct. Other than this study, there is a scarcity of data on this antigen target, which leaves the door open to many possible routes for further research. Future CAR-NK studies may further elaborate upon the clinical value of CD126 as an antigen target.

Axl and MerTK are closely related RTKs that have both been associated with drug resistance and immunological resistance in melanoma [236,237,238,239] Antibody-mediated inhibition of Axl appeared to restore patient susceptibility to BRAF inhibitors [240]. Axl inhibition was also associated with melanoma cell motility and therefore required increased security to safeguard against the risk of an invasion [241,242]. MerTK expression is upregulated in BRAFi-resistant melanoma cells and TAMs [243,244]. Correspondingly, MerTK inhibition promoted the apoptosis of melanoma cells [245], along with suppression of tumor proliferation and growth [246]. 

CD271 (nerve growth factor receptor; NGFR) is upregulated in a subset of melanoma tumors and is implicated in resistance to both drug treatments and immune responses. After exposure to differentiation antigen-specific cytotoxic T cells, melanoma cells exhibited CD271 upregulation, which may predict resistance to anti-PD-1 therapies and immune cell exclusion from tumors [247]. CD271 is also associated with melanoma intra-tumoral heterogeneity [248] and metastatic migration of tumor cells [249]. Importantly, CD271 has been linked to immune escape in multiple reports. CD271 can be induced by IFNγ, which is produced in high volumes by NK cells, thus contributing to immune escape by driving the downregulation of melanoma antigens [250,251]. If future CAR-NK studies can be designed to secrete a synthetic peptide that can inhibit CD271, this may be able to restore endogenous effector cell recruitment for concerted tumor cell elimination alongside CAR-NK cells. 

## 5. Perspectives and Future Prospects for CAR-NK Cell Therapies to Treat Melanoma

Current challenges faced by CAR-NK cell therapies include their limitation to surface antigens, neglect to structurally optimize the CAR for NK cell signal transduction, their susceptibility to immunosuppression by both the TME and malignant cells, their insufficient infiltration into tumors, and the general struggle with tumor clearance due to pro-tumoral signaling pathways that accelerate malignant growth. Future CAR-NK cell studies are likely to benefit from strengthening their cytotoxic activation as they seek out solutions to the challenges described (Figure 4).

### 5.1. Structural Optimization of the CAR to Enhance CAR-NK Cell Activation

Signal optimization is an important consideration for the optimization of CAR structures. Although TCR-T activation is generally more sensitive than CAR-T activation [252], this disparity does not necessarily carry over to CAR-NK and TCR-NK cells. NK cells are equipped with several pathways that can transduce activating signals, so it would be just as promising to try NK-specific signaling components. Presently, most CAR-NK cell constructs still use CARs designed to imitate TCR signaling [253] and may suffer from impaired activity as a result. CAR components usable in CAR-T cells may be incompatible with CAR-NK cell activity, and vice-versa. For instance, the effectiveness of 4-1BB as a costimulatory molecule in CAR-NK cells is contested, where conflicting findings hinted at an underlying incompatibility [254]; the unique significance of the adapter proteins DAP10 and DAP12 in NK cell signal transduction lends further credence to this idea [255]. CAR-NK cell therapies have been shown to perform best when equipped with components that have been tailored to NK cell signaling in studies [256,257,258]. A study on CAR-NK cell designs reported that CAR-NK cell cytotoxicity was highest for the CAR designed with an NKG2D transmembrane domain and substituting 2B4 and DAP10 as the chosen costimulatory molecules. All of these signaling components had been derived directly from NK-specific signaling components [259]. Likewise, a CAR-NK cell construct containing an NK-derived DNAM1 transmembrane domain outperformed an equivalent design bearing a T cell-specific CD28 transmembrane domain [157]. CAR-NK cell constructs possessing the NK-derived DAP12 signaling motif also outperformed equivalent constructs bearing the CD3ζ chain signaling motif [260,261], and this is a particularly noteworthy feat in that it established that TCR signaling mechanisms were not necessarily the best route to achieve NK cell activation.

### 5.2. Integration of TCR Structural Elements to Expand the Range of Targetable Antigens

Like antibodies, CAR-NK cells express a CAR that is similarly restricted to surface antigen targets. Finding ways to extend the range of potential antigen targets for CAR-NK cells has the potential to be pivotal for this prognosis. TCRs permit T cells to target intracellular antigens via MHC-I recognition [262], but despite this, engineered TCR-T cell therapies remain impractical due to their inherent vulnerability to TCR subunit mispairing between existing and transduced TCRs, which can lead to serious off-target toxicities [263,264]. NK cells do not naturally express TCRs, so the genes for a TCR can be transduced into NK cells without incurring the risks of subunit mispairing. Recently, TCR-NK cells expressing a transgenic TCR have been produced successfully [265]. A similar study reported that transgenic TCRs could be safely co-expressed alongside a CAR in CAR-T cells [211]. 

The TCR/CAR dual receptor layout would therefore permit CAR-NK cells to target the full breadth of melanoma antigens. However, this design can be consolidated into a mono-receptor TCR/CAR hybrid layout. The recent development of TCR-CARs involved the fusion of a TCR antigen recognition moiety to the intracellular CAR signaling moiety, successfully combining the strengths of both receptor types [266]. Future CAR-NK studies could further refine this design by incorporating NK-specific design elements in the intracellular CAR signaling domain. 

### 5.3. Targeting Melanoma Cell Signaling Pathways

Melanoma cells frequently exhibit mutations to driver genes enriched in several protein kinase signaling pathways that result in the aberrant hyperactivation of these pathways. BRAF mutations are the most common type of mutation in melanoma, and this can affect multiple signaling pathways within the MAPK/ERK signaling network [267,268]. Protumoral signaling pathways in melanoma are generally governed by one or more RTKs that were overexpressed frequently in melanoma cells [269]. Inhibited tumor invasion, growth, and proliferation, as well as drug and immunological resistance, have all been reported following RTK inhibition [270].

Inhibition of the signaling pathways by inhibitors used in combination with CAR-NK cell targeting specific tyrosine kinase receptors could inhibit melanoma invasion and proliferation or kill the tumor cells directly. Alternatively, CAR-NK cells could be similarly designed to secrete an inducible peptide inhibitor for a myriad of melanoma cell receptors, analogous to the TRUCK concept in CAR-T cells [173,189]. For instance, the transmembrane protein LRIG1 is expressed by NK cells [271], and the soluble ectodomain released by proteolytic cleavage (sLRIG1) has been shown to efficiently impede malignant growth in models of glioma by inhibiting several RTKs, chiefly within EGFR receptor family [179,272]. Furthermore, sLRIG1-mediated EGFR inhibition was demonstrated to inhibit melanoma growth and proliferation both in vitro [273], hinting at a favorable likelihood of similar effects in vivo. sLRIG1 is already available commercially as a reagent, so it can be inferred that DNA and peptide sequences are both already known. If sLRIG1 secretion were to be coupled to antigen recognition by the scFv of a CAR-NK cell then CAR-NK cells could potentially be used as both an effector cell and as a vehicle for localized drug delivery simultaneously. Future studies may be able to rely upon methods such as phage display [274] to elicit similarly viable peptides for use as receptor inhibitors.

### 5.4. Targeting Cell Motility to Enhance Immune Infiltration and Impair Malignant Invasion

Chemotaxis influences the activity of both effector and target cells; the motility of melanoma cells is a major driving force in metastatic migration, while NK cell motility can heuristically measure the degree of immune cell infiltration within a tumor. It follows that NK cell cytotoxicity, and by extension their clinical efficacy, can be enhanced in the presence of chemotactic cytokines and chemokines. NK cells must be able to migrate to metastatic sites if they are to eliminate metastases effectively. Chemokine receptors on NK cells such as CXCR3 enable robust NK cell chemoattraction [275]; conversely, CXCR3-deficient NK cells failed to migrate to metastatic sites [276], suggesting that increasing CXCR3 in CAR-NK cells may promote CAR-NK cells’ tumor site infiltration and enhance the CAR-NK mediated tumor clearance. Chemotactic motility can also be impacted by the expression of autophagy genes. By blocking the autophagy gene Beclin1 (BECN1) in a model of melanoma, inhibition of CCL5 was alleviated and led to inhibited growth of melanoma cells concomitantly to increased NK cell infiltration into the tumor [277]. 

### 5.5. Targeting Non-Malignant Cell Types in the TME

The assortment of cell types within the TME do not act as isolated contributors to immune escape; rather, they form a complex cell signaling network involving many different cell-cell interactions. In lieu of targeting a novel candidate melanoma surface antigen, CAR-NK cells that are alternatively redirected against antigens expressed by non-malignant TME cell types would most likely reduce the population of cells secreting immunosuppressive signals and improve the odds of preserving NK cell cytotoxicity.

CAFs may promote drug resistance and metastatic invasion in melanoma, so their destruction would likely have great clinical value. Co-culture experiments indicate that CAFs are drivers of tumor cell invasion [278]. The relationship between CAFs and RTKs expressed by melanoma cells, which mediate signaling that leads to increased MAPK activation, has also been implicated in BRAF inhibitor resistance in BRAF-mutant patients [279]. BRAF inhibitors can reprogram CAFs to drive matrix remodeling in melanoma [280], and it follows that CAFs may influence melanoma cell responsiveness to combinatorial therapy involving both BRAF and MEK inhibitors [281]. Inhibition of CAFs by β-catechin was associated with pronounced decreases in tumoral vascularization, which hints at the importance of CAFs in metastatic development and tumor progression [282]. CAFs have also been identified as mediators of immune escape as demonstrated by the impairment of CD8^+^ cytotoxic T cell activity via arginase-mediated interference with tumor cell recognition [132]. CAFs have also been reported to mediate inhibitory phenotypic shifts in NK cells [283]. Fortunately, CAFs are identifiable by a number of marker genes, including FAP [284], Col11A1 [285], and Acta2 [286], thus they could be specifically targeted by CAR-NK cell therapies in the coming years.

TAMs are another significant contributor to poor patient outcomes in melanoma. Crosstalk between CAFs and TAMs has been noted to exacerbate tumorigenesis and immune escape [287], so if CAFs are not a viable target for CAR-NK cells, then it may be worthwhile to target TAMs instead. TAMs are macrophages that have been functionally shifted to the pro-inflammatory M2 phenotype by melanoma cells [90]. M2 macrophages are known to inhibit the activity of anti-tumoral M1 macrophages [288]. In addition, TAMs express the β3-adrenoreceptor and secrete adrenomedullin, which are both associated with increased vascularization and malignant invasion in melanoma [90,289]. Therefore, the elimination of TAMs would be expected to improve immunological responsiveness against tumor cells. To date, there is at least one example of a CAR-NK cell therapy directed against TAMs, in which anti-CSF1R CAR-NK92MI and CAR-T cells were developed and exhibited strong anti-TAM activity [290]. Future CAR-NK cell studies may be able to elicit comparable successes by targeting other known TAM markers such as CCL5, CD163, or AHR [291].

Finally, T_reg_ cells are secretory T cells that produce a plethora of immunosuppressive signaling molecules that protect melanoma tumors from immune destruction [292,293,294,295]. Importantly, T_reg_ secretion of perforin and granzyme-B are directly associated with NK cell inhibition [296]. A decrease in the T_reg_ cell population would correspond to a lessened immunosuppressive environment for CAR-NK cells. Recently, a CAR-NK cell study targeting CD25 was published to target T_reg_ cells that were newly reprogrammed to a pro-tumoral state [297]. Other markers such as galectin-9 [298] and PD-L1/L2 [299,300] may also be potential targets for CAR-NK cell therapies that are targeted against T_reg_ cells.

### 5.6. Incoporation with Deletion of Inhibitory Receptor Using Innovated Methods

NK cell inhibitory receptors are potential immune checkpoint therapeutic targets [32]. Blocking these inhibitory receptors could enhance the capability of NK cells to killing the tumor cells [115,200]. CRISPR/Cas9 technology has enabled site-specific inhibitory receptor gene deletion. Recent study has demonstrated that combining adeno-associated viral (AAV) system with CRISPR/Cas9 technology could successfully insert the gene into NK cells with site-specifically [301]. This innovative method may be the most promising approach to develop next generation CAR-NK cells. 

### 5.7. Clinical Consideration of Pharmaceutical Impacts on CAR-NK Cell Efficacy

As a final consideration, it is imperative that future CAR-NK cell therapies be optimized to operate in tandem with drug therapies. NK cells are directly responsible for the clinical successes of BRAF inhibitors [302] and acquired resistance to BRAF inhibitors has been associated with increased melanoma cell susceptibility to NK cell lysis [303]. If subsequent studies can clarify the potential risk of BRAF inhibitor cytotoxicity towards CAR-NK cells, the clinical versatility of cell immunotherapies may be confirmed. BRAF inhibitors may begin to be administered as an adjuvant to CAR-NK cell therapies, or perhaps CAR-NK cells may become a follow-up therapy to BRAF inhibitors by capitalizing on the phenotypic shifts associated with acquired drug resistance [304]. Future studies that explore the dynamics between NK cells and other drug therapies would shed light on the compatibility of CAR-NK cells with other adjuvant therapies. Promising clinical implications may be revealed by these studies, as well as potentially unfavorable interactions that clinicians should consider when formulating treatment plans for melanoma patients.

## 6. Conclusions

The rapid development of drug resistance contributes significantly to the ongoing clinical challenges of melanoma treatment. The persistently high mortality rate of melanoma patients, especially those with metastatic melanoma, is indicative of an ongoing, unmet need for a sufficiently efficacious treatment. CAR-T cell therapies have marked a significant milestone for improving patient outcomes for hematological malignancies, but the accompanying risks and limitations of CAR-T cell therapies suggest that CAR-NK cells may be a more suitable alternative immunotherapy. NK cells are mechanically distinct from T cells, which enables them to continue to target tumor cells even in cases of antigen loss, but this comes at the cost of increased sensitivity to inhibitory signals secreted within the TME. As CAR-NK cell therapies continue to grow and improve, the impact of these limitations will be lessened with time. There is a favorable degree of flexibility for the myriad of therapeutic strategies that may be useful for future studies on CAR-NK cells. Further research will provide greater insights into the clinical utility of CAR-NK cells for the treatment of melanoma.

## Figures and Tables

**Figure 1 cells-12-02750-f001:**
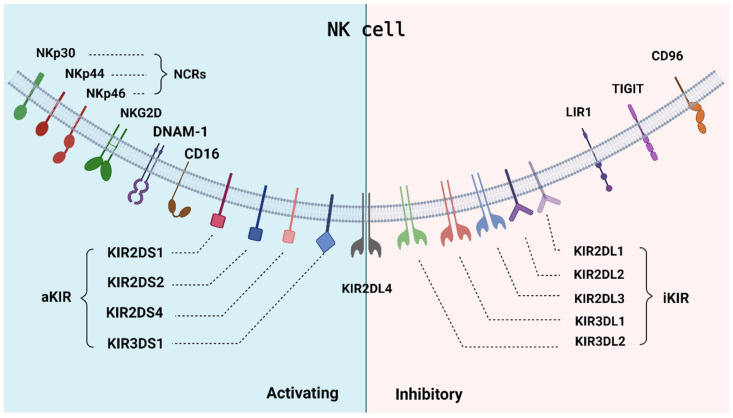
Natural killer cell signaling receptor repertoire. Natural killer cells are equipped with a diverse assortment of activating and inhibitory signaling receptors that mediate NK cell activity through combined signal propagation. Activating receptors, including NCRs, NKG2D, DNAM-1, CD16, and aKIRs transmit the signals to promote NK cell activation and cytotoxicity, while inhibitory receptors, including CD96, TIGIT, LIR1 and iKIRs, induce the inhibition signaling to block the NK cell activation. KIR2DL4 could introduce activating or inhibitory signaling into NK cells based on its content and ligands.

**Figure 2 cells-12-02750-f002:**
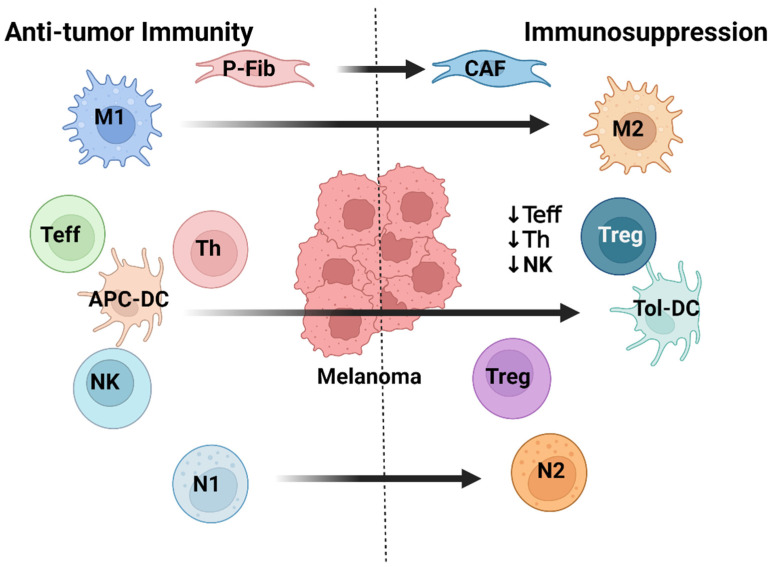
Functional shifts between anti-tumor immunity and immunosuppression within the tumor microenvironment. Several non-malignant immune and stromal cells undergo phenotypic alterations through direct cell-cell interactions with melanoma cells and through the influence of chemical conditions within the tumor microenvironment. M1: phagocytic macrophage phenotype. M2: pro-inflammatory macrophage phenotype. N1: cytotoxic neutrophil phenotype. N2: pro-inflammatory neutrophil phenotype. APC-DC: functional antigen-presenting dendritic cell. Tol-DC: dysfunctional tolerogenic dendritic cell. Teff: CD8+ cytotoxic T cell. Th: CD4+ pro-cytotoxic helper T cell. Treg: CD4+ immunosuppressive regulatory T cell. NK: natural killer cell. P-Fib: primary fibroblast. CAF: cancer-associated fibroblast.

**Figure 3 cells-12-02750-f003:**
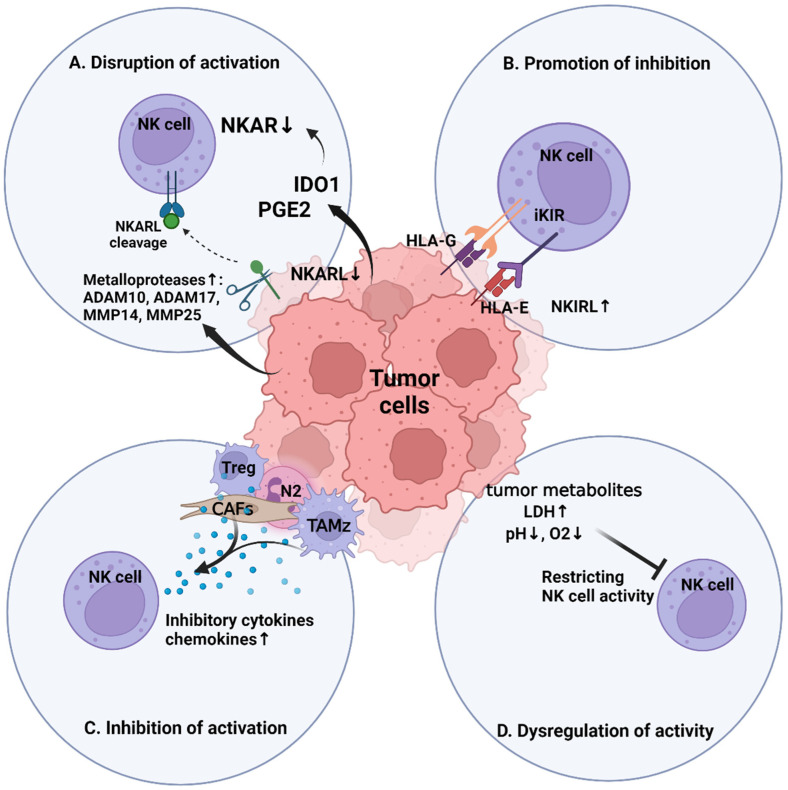
The four main mechanisms of immune escape. (**A**) Disruption of NK cell activation through at least three mechanisms: Tumor cells can express molecules such as IDO1 and PGE2 to mediate the inhibition of NKAR expression in NK cells. Also, tumor cells often downregulate the NKAR ligand expression on tumor cell surfaces or/and shed their NKAR ligands proteolytically. (**B**) Promotion of NK-inhibiting receptor-ligand interactions via upregulation of inhibitory ligand expression on melanoma cells. (**C**) Immune escape assisted by non-malignant constituent cells within the tumor microenvironment through secreted immunosuppressive cytokines and chemokines. (**D**) Immunological dysfunction resulting from hypoxic and acidic chemical conditions within the tumor microenvironment.

**Figure 4 cells-12-02750-f004:**
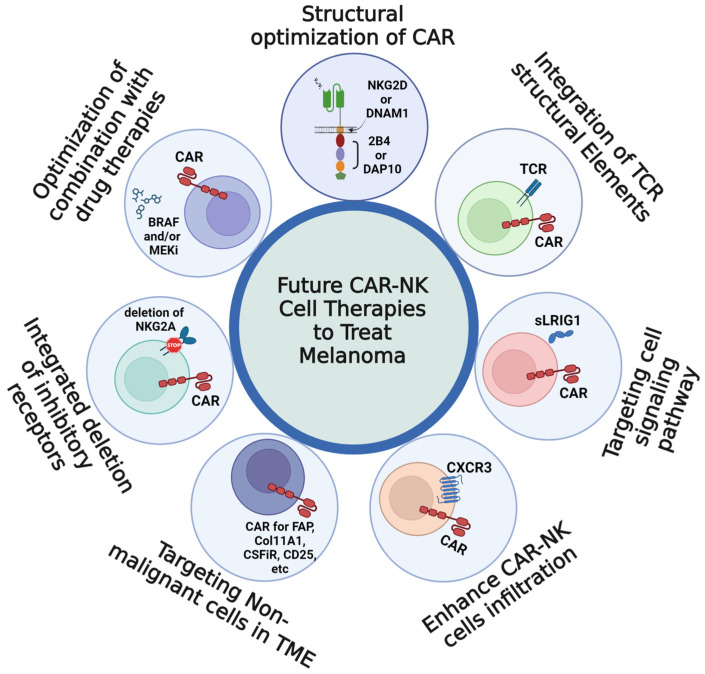
A diagram of future CAR-NK cell development for treatment of melanoma. These studies will focus strengthening CAR-NK cytotoxic activation for maximized to killing the metastatic and drug resistant melanoma, including structural optimization of CAR with NKG2D or DNAM1 transmembrane domain and 2B4 or DAP10 intracellular signaling domain, integration of TCR structural elements in CAR-NK cells, targeting receptor tyrosine kinase signaling with sLRIG1, integration of CXCR3 to enhance CAR-NK cells infiltration, targeting markers of non-malignant cells in TME, corporation of blocking NK cell inhibitory receptors, as well as optimization with drug therapies.

**Table 1 cells-12-02750-t001:** Current pre-clinical and clinical CAR-NK cell studies relevant to melanoma.

Target	Study Type	Clinical Trial Status	Cancer Type	Reference
5T4	Clinical	Active/Recruiting	Solid tumors	NCT05194709
B7-H3	Preclinical	-	Non-small cell lung cancer	[161]
B7-H6	Preclinical	-	Solid tumors	[162]
CD24	Preclinical	-	Ovarian cancer	[163]
CD276	Preclinical	-	Solid tumors	[164]
CD44	Preclinical	-	Ovarian cancer	[165]
CLDN6	Clinical	Active/Recruiting	Solid tumors	[166]
c-Met	Preclinical	-	Lung adenocarcinoma	[160]
EGFR/EGFRvIII	Preclinical	-	Glioblastoma	[167]
		-	Glioblastoma	[168]
		-	Glioblastoma	[169]
		-	Glioblastoma	[170]
EpCAM	Preclinical	-	Solid tumors	[171]
GD2	Preclinical	-	GD2+ solid tumors	[172]
		-	GD2+ solid tumors	[173]
		-	Ewing sarcoma	[174]
gp100	Preclinical	-	melanoma	[175]
HER2	Preclinical	-	Rhabdomyosarcoma	[146]
		-	Glioblastoma	[176]
		-	Solid tumors	[177]
		-	Rhabdomyosarcoma	[178]
Mesothelin	Preclinical	-	Gastric cancer	[158]
MUC1	Clinical	Completed	Solid tumors	[179]
NKG2DL	Clinical	Active/Recruiting	Solid tumors	NCT05528341
		Recruiting	Solid tumors	NCT03415100
ROBO1	Clinical	Unknown	Solid tumors	NCT03940820
VEGFR2	Preclinical	-	Solid tumors	[180]

**Table 2 cells-12-02750-t002:** Potential targets for future CAR-NK cell immunotherapies for melanoma based on CAR-T cell preclinical and clinical studies.

Target	Study Type	Clinical Trial Status	Cancer Type	Reference
AXL	Preclinical	-	Triple-negative breast cancers	[202]
B7-H3	Clinical	Active, Not Recruiting	Solid tumors	NCT04483778
		Recruiting	Solid tumors	NCT04897321
		Recruiting	Solid tumors	NCT05190185
	Preclinical	-	Solid tumors	[194]
		-	Glioblastoma	[197]
		-	Solid tumors	[198]
		-	Prostate cancer	[199]
		-	Solid tumors	[203]
B7-H3/CD70	Preclinical	-	Solid tumors	[204]
CD126	Preclinical	-	Solid tumors	[205]
CD16	Preclinical	-	Solid tumors	[206]
CD70	Clinical	Recruiting	Solid tumors	NCT02830724
	Preclinical	-	CD70+ cancers	[207]
		-	CD70+ cancers	[208]
c-Met	Clinical	Completed	Melanoma, breast carcinoma	[209]
	Preclinical	-	Gastric cancer	[210]
CSPG4	Preclinical	-	Solid tumors	[211]
		-	Solid tumors	[212]
		-	Melanoma	[213]
GAS6	Preclinical	-	Pancreatic cancer	[214]
GD2	Clinical	Completed	Melanoma	[186,215]
		Completed	Solid tumors	[216]
		Completed	Neuroblastoma	[217]
		Recruiting	Solid tumors	NCT03635632
		Completed	Solid tumors	NCT02107963
	Preclinical	-	Lung cancer	[185]
		-	Solid tumors	[189]
		-	Solid tumors	[215]
GD3	Preclinical	-	Melanoma	[218]
		-	Melanoma	[219]
HER2	Preclinical	-	HER2+ sarcomas	[29]
		-	HER2+ melanoma subtypes	[220]
IL13Ra2	Clinical	Recruiting	Solid tumors	NCT04119024
Multiple targets (NY-ESO-1, DR5, EGFRvIII, c-Met)	Clinical	Unknown	Solid tumors	NCT03638206
SLC45A2	Preclinical	-	Melanoma	[221]
TRP-1	Preclinical	-	Melanoma	[222]
VEGFR2	Clinical	Terminated	Solid tumors	NCT01218867
	Preclinical	-	Solid tumors	[180]
		-	Solid tumors	[223]
αvβ3	Preclinical	-	Glioma and glioblastoma	[224]
		-	Advanced cancers	[225]

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
