# Peer review of "CAR NK Cell Therapy for the Treatment of Metastatic Melanoma: Potential & Prospects"

_cells, 2023, doi:10.3390/cells12232750_

Round 1

Reviewer 1 Report

Comments and Suggestions for Authors

In this review Hibler et al review the current status of CAR-NK cell immunotherapies for the treatment of melanoma, highlighting the challenges in targeting solid tumours and the potential targets that may be used to direct NK cells successfully towards melanoma. Please find below my specific questions and comments for this manuscript:

Line 16: The phrasing “Natural killer (NK) cells for CAR natural killer (CAR-NK)….” is a little bit confusing and could be rephrased.

Lines 59-66: The review states that the application of CAR T cells in solid tumours is limited by toxicities eg. GvHD, cytokine release syndrome etc. however these all constraints that also hamper the use of CAR T cells in haematological malignancies. Additional text discussing the additional barriers to CAR therapy in solid tumours such as cell trafficking to tumours, penetrance of solid tumours, tumour heterogeneity etc. should be included here. In addition, in this section of text there is mention of the problems of downregulation of MHC-I and reduced T cell cytotoxicity which is a problem but not necessarily a problem specific to CAR T cells that don’t require MHC-I to be activated. I wonder if this is supposed to refer to additional mechanisms that the CAR T cells could eliminate the cancer cells and additional context is required to support this statement. 

Lines 72-74: Regarding the lack of GvHD and reduced toxicity with CAR-NK cells references should be included.

Line 74-75: The reduced risks is not necessarily why NK cell therapy can be ‘off the shelf’ it is because NK cells to mediate GvHD (which does relate to the toxicity) but this sentence could be rephrased to explain this important aspect of NK cell biology and thus why they can be used ‘off the shelf’.

Line 83: NK cells can also be found in the uterus and mucosal tissues and most likely other tissues as well. Perhaps this sentence could be made less specific. 

Line 87: CD56 expressing level should be CD56 expression level.

Line 89: FCy should be lower case c (Fcy).

Line 91: matured CD56dim NK cells should be mature CD56dim NK cells.

Lines 125-130: It’s very important to point out that NKAR do not induce cytotoxicity alone apart from CD16 as stated but this feels out of place at this point in the paragraph (which is then later followed by aKIR). I would think about reordering this section to make it flow better for the reader.

Line 138: CD94:NKG2C heterodimer is mentioned but hasn’t been mentioned before. It would be helpful for the review to include NKG2C as another example of an NKAR.

Line 149-150: More detail about the different HLA that are recognised by iKIR could be included. It reads as if KIR3DL1/2 can recognise all HLA-A and -B alleles when it is only a subset.

Line 152: Perhaps aKIR would be better here than NKAR.

Line 173: ‘transduce the signalling to promote NK cell activation’ needs to be rephrased. 

Line 175: This could be removed as it doesn’t seem essential to the figure. 

Section 2.3: The authors could consider moving this section earlier to give more context to the section on receptors as this section describes the importance of the balance of activating and inhibitory signals and the recognition of self-HLA.

Lines 197-200: This sentence is quite confusing and difficult to follow what is important for the reader to understand. 

Line 230: type with an extra : 

Line 237: Change could to can

Line 254: Change downregulating to downregulation or remove the of

Line 303: Rephrase ‘levels can also prevent TAMs from activating against tumour cells’ as activating against is not clear. 

Lines 325-331: This final paragraph in section 3 requires either a summary of the state of NK cells in the TME which leading onto why CAR-NK cells may offer a potential solution or something equivalent because on its own there is little context for it. 

Lines 361 – 383: In this paragraph the authors could consider adding more context to these different types of 4th generation CAR constructs. For example introduce AND/OR-gated CARs.

Line 393: Change cell line NK cells to NK cell lines

Lines 400-406: It is not clear to the reader why there would be a need to minimise heterogeneity in the NK cell population. Additional context is required here. Presumably this is referring to a population of highly cytotoxic NK cells but this needs to be made clear. In addition, this is mention of an unfavourable heterogenous population but what does that refer to.

Line 407: Spell out iPSC

Line 422: In table 1 is the reference 259 for MUCI trial the correct reference because that reference in the reference section is focused on B7-H6. In addition, the clinical trials in this table aren’t discussed in the text below. Perhaps the status of these trials should be summarised and what solid tumour they are targeting. It does look some of it is discussed in the next section 4.4 but it would be more relevant to discuss it earlier.

Lines 468: This table could be improved by adding a column describing the cancer that is currently being targeted by these CAR NK cells. Also some of these targets are included in the previous table. What is the difference?

Lines 493 – 495: While it is true that the reduced levels of toxicity of CAR NK cells make them preferable additional context is required as to why this is important in brain metastases. Have the CAR T cell products reported toxicities in this setting?

Sections 4.3 and 4.4: Overall this section of the review is quite challenging to follow. There is a lot of information and descriptions of different targets that have been used in the field of CAR NK cells to treat solid tumours. However, there is a lack of the authors opinions throughout this section. Which targets do they see as being the most promising? What are some of the issues with these targets? By adding in their unique perspective throughout this section will help steer the reader and elevate the review. 

Line 556: Wealth of publications could be changed to studies and references are required.

Line 568: Needs rephrasing as it is not clear.

Line 570: Change his to this.

Section 5: A picture/diagram summarising these different points would be very useful for the reader.

Comments on the Quality of English Language

See comments above. 

Author Response

Specific responses to the points raised by the reviewers are as follows: Our responses are in bold courier new for easier visualization.

Reviewers' comments: 

Reviewer #1

In this review Hibler et al review the current status of CAR-NK cell immunotherapies for the treatment of melanoma, highlighting the challenges in targeting solid tumours and the potential targets that may be used to direct NK cells successfully towards melanoma. Please find below my specific questions and comments for this manuscript:   

We thank the reviewer for so carefully reading our manuscript and making many substantive comments.

Line 16: The phrasing “Natural killer (NK) cells for CAR natural killer (CAR-NK)….” is a little bit confusing and could be rephrased.

The phrasing has been updated for clarity.

Lines 59-66: The review states that the application of CAR T cells in solid tumours is limited by toxicities eg. GvHD, cytokine release syndrome etc. however these all constraints that also hamper the use of CAR T cells in haematological malignancies. Additional text discussing the additional barriers to CAR therapy in solid tumours such as cell trafficking to tumours, penetrance of solid tumours, tumour heterogeneity etc. should be included here. In addition, in this section of text there is mention of the problems of downregulation of MHC-I and reduced T cell cytotoxicity which is a problem but not necessarily a problem specific to CAR T cells that don’t require MHC-I to be activated. I wonder if this is supposed to refer to additional mechanisms that the CAR T cells could eliminate the cancer cells and additional context is required to support this statement. 

We appreciate the reviewer’s points. The paragraph has been revised for clarity. Tumor heterogeneity has been included explicitly in the discussion and trafficking to tumors has been included implicitly when describing the reduced risk of on-target, off-tumor toxicities. This section is intended to briefly highlight the advantages of NK cells as an alternative effector cell for inducing CAR expression, but tumor penetrance similarly affects CAR-NK cells. A comprehensive comparison of the advantages and disadvantages of CAR-NK versus CAR-T therapies is outside the scope of this review.

Instead, we have discussed chemotaxis in the latter sections on proposed design innovations.

Lines 72-74: Regarding the lack of GvHD and reduced toxicity with CAR-NK cells references should be included.

Multiple appropriate references have been added. 

Line 74-75: The reduced risks is not necessarily why NK cell therapy can be ‘off the shelf’ it is because NK cells to mediate GvHD (which does relate to the toxicity) but this sentence could be rephrased to explain this important aspect of NK cell biology and thus why they can be used ‘off the shelf’.

I appreciate the reviewer for the suggestion. We have updated the paragraph for explanation: “unlike T cells, NK cell activation does not require HLA matching for antigen recognition and its cytotoxicity mechanisms are regulated by multiple activating and inhibitory receptors. This unique aspect of NK cell activity allows GvHD to be avoided in allogeneic NK cell therapies, which in turn enables allogeneic NK cells to be made readily available as an “off-the-shelf” treatment.”

Line 83: NK cells can also be found in the uterus and mucosal tissues and most likely other tissues as well. Perhaps this sentence could be made less specific. 

The phrasing has been revised as recommended. 

Line 87: CD56 expressing level should be CD56 expression level.

The phrasing has been revised as recommended. 

Line 89: FCy should be lower case c (Fcy).

The letter case has been revised as recommended.   

Line 91: matured CD56dim NK cells should be mature CD56dim NK cells.

The phrasing has been revised as recommended. 

Lines 125-130: It’s very important to point out that NKAR do not induce cytotoxicity alone apart from CD16 as stated but this feels out of place at this point in the paragraph (which is then later followed by aKIR). I would think about reordering this section to make it flow better for the reader.

The paragraphs were reordered for flow as recommended.

Line 138: CD94:NKG2C heterodimer is mentioned but hasn’t been mentioned before. It would be helpful for the review to include NKG2C as another example of an NKAR.

We thank the reviewer for pointing it out. The NKG2C has been included in NKAR.

Line 149-150: More detail about the different HLA that are recognised by iKIR could be included. It reads as if KIR3DL1/2 can recognise all HLA-A and -B alleles when it is only a subset.

The section has been revised for clarity with some details added.

Line 152: Perhaps aKIR would be better here than NKAR.

We thank the reviewer for pointing it out. NKAR has been revised to aKIR.

Line 173: ‘transduce the signalling to promote NK cell activation’ needs to be rephrased.

The phrasing has been revised as recommended.

Line 175: This could be removed as it doesn’t seem essential to the figure. 

It was removed as recommended.

Section 2.3: The authors could consider moving this section earlier to give more context to the section on receptors as this section describes the importance of the balance of activating and inhibitory signals and the recognition of self-HLA.

The preceding sections had described NK receptors, so section 2.3 contextualizes NK receptor behavior in biological and clinical contexts. We believe it is best to keep this section as it is.

Lines 197-200: This sentence is quite confusing and difficult to follow what is important for the reader to understand.

This sentence has been revised for clarity as recommended.

Line 230: type with an extra :

The correction has been made as recommended.

Line 237: Change could to can

Revision has been made as recommended.

Line 254: Change downregulating to downregulation or remove the of

Revision has been made as recommended.

Line 303: Rephrase ‘levels can also prevent TAMs from activating against tumour cells’ as activating against is not clear. 

The phrasing has been adjusted for clarity, as recommended.

Lines 325-331: This final paragraph in section 3 requires either a summary of the state of NK cells in the TME which leading onto why CAR-NK cells may offer a potential solution or something equivalent because on its own there is little context for it. 

Since the immunosuppressive effects have already been described, we feel that it would be more practical to revise phrasing to explicitly connect CAR expression to counteracting TME immunosuppression. This section has been revised accordingly.

Lines 361 – 383: In this paragraph the authors could consider adding more context to these different types of 4th generation CAR constructs. For example introduce AND/OR-gated CARs.

I appreciate the reviewer for the suggestion. The content of AND/OR-gated CARs has been updated.

Line 393: Change cell line NK cells to NK cell lines

The phrasing has been revised as recommended.

Lines 400-406: It is not clear to the reader why there would be a need to minimise heterogeneity in the NK cell population. Additional context is required here. Presumably this is referring to a population of highly cytotoxic NK cells but this needs to be made clear. In addition, this is mention of an unfavourable heterogenous population but what does that refer to.

The phrasing has been revised for clarity. Since we have clarified the nature of the referenced heterogeneity, this should address the second comment as well.

Line 407: Spell out iPSC

iPSC is the abbreviation of ‘induced pluripotent stem cell; The words have been updated.

Line 422: In table 1 is the reference 259 for MUCI trial the correct reference because that reference in the reference section is focused on B7-H6. In addition, the clinical trials in this table aren’t discussed in the text below. Perhaps the status of these trials should be summarised and what solid tumour they are targeting. It does look some of it is discussed in the next section 4.4 but it would be more relevant to discuss it earlier.

We apologize for the confusion. Our table has been updated accordingly.

Lines 468: This table could be improved by adding a column describing the cancer that is currently being targeted by these CAR NK cells. Also some of these targets are included in the previous table. What is the difference?

The table has been updated accordingly. For some of these targets in both tables, CAR-T tends to offer more clinical trial data, whereas CAR-NK cell studies are still largely in the pre-clinical phase.

Lines 493 – 495: While it is true that the reduced levels of toxicity of CAR NK cells make them preferable additional context is required as to why this is important in brain metastases. Have the CAR T cell products reported toxicities in this setting?

Yes. We have mentioned this a few times (e.g. ICANS). Another reviewer has been critical of “redundancies” so we unfortunately may not be at liberty to fully clarify in this regard.

Sections 4.3 and 4.4: Overall this section of the review is quite challenging to follow. There is a lot of information and descriptions of different targets that have been used in the field of CAR NK cells to treat solid tumours. However, there is a lack of the authors opinions throughout this section. Which targets do they see as being the most promising? What are some of the issues with these targets? By adding in their unique perspective throughout this section will help steer the reader and elevate the review. 

We have selected a handful of promising targets to discuss. We hope that readers can infer that these are the notable examples.

Line 556: Wealth of publications could be changed to studies and references are required.

Revision has been implemented as recommended.

Line 568: Needs rephrasing as it is not clear.

Phrasing has been revised as recommended.

Line 570: Change his to this.

The correction has been made as recommended.

Section 5: A picture/diagram summarising these different points would be very useful for the reader.

We thank the reviewer for the great suggestions. A diagram of future CAR-NK cell development for the treatment of melanoma has been summarized in Figure 4.  

Reviewer 2 Report

Comments and Suggestions for Authors

In comparison to CAR-T therapy, several recent studies from different laboratories agree that CAR-NK cell therapy is safer with a lower chance of negative immune side effects and shows more efficient antitumor activity. Furthermore, CAR-NK cell therapies for solid tumors and hematological malignancies have shown promising results in clinical trials.

The manuscript/review by Winston Hibler and colleagues titled “CAR NK Cell Therapy in the Treatment of Metastatic Melanoma: Potential & Prospects” is an exhaustive, well-structured and well documented review of the literature with an up-to-date bibliography. All aspects of this novel immunotherapy approach are thoroughly discussed with a focus on the context of melanoma, including the biology and function of NK cells and the tumor microenvironment. Finally, the limitations on both obtaining and using CAR NK cell therapies to treat melanoma are clearly highlighted and discussed.

Minor comments

The title is somehow confusing since melanoma has not yet been specifically targeted using CAR-NK in clinical trials probably due to the absence of a specific targetable antigen together with the complexity of the tumor microenvironment of melanoma, I'd suggest qualifying the title:

The Use of CAR NK Cell Therapy in the Treatment of Metastatic Melanoma: Potential & Prospects

Author Response

We appreciate the reviewer for all the positive comments and suggestions. We have implemented the title revision accordingly.

Reviewer 3 Report

Comments and Suggestions for Authors

This review manuscript of Winston Hibleret al. is devoted to possibilities of melanoma therapy based on CAR-NK cells. The topic is of great interest and importance. The authors have collected a large amount of diverse information more or less related to the topic. Some sections of the manuscript are well written. However, in addition to aspects directly related to CAR-NK cell applications for melanoma treatment, the manuscript contains a lot of redundant information on some issues. In particular, the authors described in separate paragraphs the textbook data about NK сell activating and inhibitory receptors and the role of NK cell cytotoxicity for tumor cell clearance. Such presentation of information does not favor for keeping the focus on the exact topic of the review. Moreover, this information is not always accurate. For example, KLRD1 (CD94) is designated as an inhibitory receptor, although CD94, as a subunit, can be part of both the inhibitory (NKG2A) and activating (NKG2C) receptors. Another issue is the manuscript contains many links to reviews, not to experimental articles, in the reference list. It also contains redundant and not always actual abbreviations, which never found further in the text (for example, LIgRs, KLRs).

I would recommend the authors to reconsider the necessity of some parts of the manuscript or to reduce significantly some paragraphs and check carefully all information presented.     

In its present form, despite the importance of the field of investigation, I would not recommend this manuscript for publication in Cells.

Author Response

We appreciate the reviewer's positive comments. LlgRs was a typing error of LILRs (leukocyte immunoglobulin-like receptors), correction has been made to the manuscript. KLRs is the abbreviation of Killer cell lectin-like receptors.

We also thank the reviewer for pointing out other issues and like to discuss the following:

Regarding the claim of redundancy: NK cell activating and inhibitory receptors are described in separate paragraphs to provide an overview of NK cell signaling diversity. The section on NK cell significance to tumor clearance then adds a clinical connection to NK cell activity and function, thus this section should not be discounted as a redundancy. The separation discussions of activating and inhibitory receptors provide the background information needed for later sections: the activating receptor repertoire is described to provide context for the discussion of the structural innovations to the CAR, while the inhibitory receptor repertoire is described to provide context for potential target genes to be excised when creating the CAR. This refers to a newly added section on innovations to CAR transduction methods that can utilize CRISPR to delete an inhibitory receptor gene concomitant to AAV-mediated transfection. Both sections are therefore non-redundant.

Regarding the accuracy of claims: CD94:NKG2C had already been acknowledged as an activating receptor in the preceding section, so we are not quite sure why this was cited as a supposed correction to the material. CD94 was discussed in the section on inhibitory receptors because CD94:NKG2A has a 6-fold greater binding affinity to HLA-E than CD:94NKG2C, making the CD94:NKG2A receptor substantially more relevant to the clinical context of cancer. Clarifying the matter was accomplished with minimal revisions.

Regarding the use of literature reviews: We agree that experimental articles are very important, but due to space limitations, we could not cite all experimental articles. We apologize to all authors whose work could not be cited in our manuscript. We would also like to point out that literature reviews are published in the same journals as experimental articles and undergo equivalent processes of peer review, thus the scientific merit of literature reviews should not be overlooked. Literature review citations enable readers to be connected to consolidated sources of more information.

Regarding abbreviations: The abbreviation of scientific names is commonly and often used in scientific literature due to it can save space and avoid distracting the reader. We are sorry for one that ‘LlgRs’ was a typing error of LILRs (leukocyte immunoglobulin-like receptors). We have made that correction to our revised manuscript.

Regarding the recommendation: Based on all reviewers’ comments and suggestions, we have made significant revision of our manuscript, we hope the revised manuscript will be a satisfactory compromise.

Reviewer 4 Report

Comments and Suggestions for Authors

The review submitted by Hibler et al. it is exhaustive and well articulated in all its parts. The figures are also well drawn and help in understanding the text.

In my opinion there are the following minor points to correct:

1. keywords are missing on the first page

2. the description of the pre-clinical and clinical studies concerning the use of CAR-NK cells in melanoma is exhaustive, but I did not understand why in tables 1 and 2 the studies described in the text are mostly not reported  as well as those included in the tables are not considered in the text.

Author Response

Specific responses to the points raised by the reviewers are as follows: Our responses are in bold courier new for easier visualization.

Reviewer #4

The review submitted by Hibler et al. it is exhaustive and well articulated in all its parts. The figures are also well drawn and help in understanding the text.

We appreciate the reviewer for the positive comments.

In my opinion there are the following minor points to correct:

  1. keywords are missing on the first page

Keywords have been added to the manuscript.

  1. the description of the pre-clinical and clinical studies concerning the use of CAR-NK cells in melanoma is exhaustive, but I did not understand why in tables 1 and 2 the studies described in the text are mostly not reported as well as those included in the tables are not considered in the text.

First, we limited our tables to CAR-NK/CAR-T cell studies targeting antigens that can or could work in melanoma, thus hematological applications were excluded as they are outside the scope of this review. Second, we wanted to highlight a select few illustrative examples of promising antigen targets; if we were to fully explore every target antigen listed in the tables, we would inadvertently drift outside of the scope of this review by excessively elaborating upon topics that are but tangentially relevant to the final sections that contain the most important topics. Third, we excluded pre-clinical studies that did not explicitly use CAR-NK cells from Table 1 (i.e. studies that examined an antigen’s expression but did not use CAR-NK cells to target it). For the CAR-T cell table, redundant

We have additionally reviewed the paper and made sure that the studies that were not screened out by the three criteria listed above are included in the tables.

Round 2

Reviewer 1 Report

Comments and Suggestions for Authors

Thank you for your revised manuscript including your new figure 4 which is a great summary picture.

Reviewer 3 Report

Comments and Suggestions for Authors

I am satisfied with the compromise revised version of the manuscript.